# The Attenuation of Insulin/IGF-1 Signaling Pathway Plays a Crucial Role in the Myo-Inositol-Alleviated Aging in *Caenorhabditis elegans*

**DOI:** 10.3390/ijms24076194

**Published:** 2023-03-24

**Authors:** Nae-Cherng Yang, Chia-Yu Chin, Ya-Xin Zheng, Inn Lee

**Affiliations:** 1Department of Nutrition, Chung Shan Medical University, Taichung 402, Taiwan; 2Department of Nutrition, Chung Shan Medical University Hospital, Taichung 402, Taiwan

**Keywords:** myo-inositol, longevity, PI3K inhibitor, insulin/insulin-like growth factor 1 signaling pathway, *C. elegans*

## Abstract

Myo-Inositol (MI) has been shown to alleviate aging in *Caenorhabditis (C). elegans.* However, the mechanism by which MI alleviates aging remains unclear. In this study, we investigate whether MI can modulate the PI3K so as to attenuate the insulin/IGF-1 signaling (IIS) pathway and exert the longevity effect. The wild-type *C. elegans* and two mutants of AKT-1 and DAF-16 were used to explore the mechanism of MI so as to extend the lifespan, as well as to improve the health indexes of pharyngeal pumping and body bend, and an aging marker of autofluorescence in the *C. elegans*. We confirmed that MI could significantly extend the lifespan of *C. elegans*. MI also ameliorated the pharyngeal pumping and body bend and decreased autofluorescence. We further adopted the approach to reveal the loss-of-function mutants to find the signaling mechanism of MI. The functions of the lifespan-extending, health-improving, and autofluorescence-decreasing effects of MI disappeared in the AKT-1 and DAF-16 mutants. MI could also induce the nuclear localization of the DAF-16. Importantly, we found that MI could dramatically inhibit the phosphoinositide 3-kinase (PI3K) activity in a dose-dependent manner with an IC50 of 90.2 μM for the p110α isoform of the PI3K and 21.7 μM for the p110β. In addition, the downregulation of the PI3K expression and the inhibition of the AKT phosphorylation by MI was also obtained. All these results demonstrate that MI can inhibit the PI3K activity and downregulate the PI3K expression, and the attenuation of the IIS pathway plays a crucial role for MI in alleviating aging in *C. elegans*.

## 1. Introduction

The insulin/insulin-like growth factor (IGF)-1 signaling (IIS) pathway regulates aging in many organisms, ranging from simple invertebrates to humans [1]. In the *C. elegans*, the IIS pathway is regulated by insulin-like peptide ligands that bind to the insulin/IGF-1 transmembrane receptor (IGFR) ortholog DAF-2. The DAF-2 controls the activity of a conserved phosphoinositide 3-kinase (PI3K)/AKT kinase cascade as a consequence of the inhibition of the class O of the Forkhead box transcription factors (FOXO) ortholog DAF-16, thereby accelerating aging in the *C. elegans* [2,3,4,5,6]. Thus, the attenuation of the IIS pathway is one of the main mechanisms to induce longevity in *C. elegans* [3,4,7,8]. For example, a PI3K inhibitor, LY-294002, has the ability to extend the lifespan of *C. elegans* by inhibiting the PI3K activity and is considered to have the potential as a caloric restriction mimetic to slow aging [9]. It is known that myo-inositol (MI) can downregulate the expression of PI3K [10,11,12]. Thus, we are interested in revealing whether MI can slow aging by downregulating the PI3K expression and thereby attenuating the IIS pathway.

Inositol is a sugar alcohol with a six-carbon ring structure. Inositol has nine isomers (scyllo-inositol, muco-inositol, D-chiro-inositol, L-chiro-inositol, neo-inositol, allo-inositol, epi-inositol, cis-inositol, and MI), of which MI is the major form of inositol in nature and in the human body [13,14,15,16]. MI deficiency may result in atherosclerosis, increased blood cholesterol levels, skin disorders, declining brain function, constipation, alopecia, and eczema [17]. MI also has many beneficial effects against human diseases, including polycystic ovary syndrome (PCOS), diabetes, psychiatric dysfunctions, oligoasthenoteratospermia and cancer [13,14,15,18,19,20,21]. In 2011, the administration of D-chiro-inositol was demonstrated to extend the lifespan of *Drosophila melanogaster* via the activation of FOXO [22]. However, the longevity effect and mechanism of MI remain unclear. Interestingly, a study in the literature recently reported that MI could alleviate the aging of *C. elegans* [23]. However, they reported that MI extended the lifespan was an AKT and DAF-16 independent in the *C. elegans*, and they found that MI can alleviate aging by activating the DAF-18/PTEN and then act through the mitophagy regulator PTEN-induced kinase-1 (PINK-1) to regulation mitophagy [23]. The AKT and DAF-16 independence means that the IIS pathway is not attenuated by the MI. However, they adversely found that the level of phosphatidylinositol 4,5-bisphosphate (PIP2) was increased because the phosphatidylinositol (3,4,5)-trisphosphate (PIP3) was turned into PIP2 by the increased PTEN expression. The turning of PIP3 into PIP2 means, theoretically, that the IIS pathway would be attenuated. Reviewing the study [23], the authors did not completely rule out the interference of the osmotic pressure on the experimental results. The authors used 50 mM of MI for their study so as to diminish the interference of the osmotic pressure. However, the 50 mM of MI may still not completely rule out the interference of the osmotic pressure, as a paper reported that the osmotic pressure caused by 55 mM of sorbitol could significantly extend the lifespan of *C. elegans* [24]. Thus, the mechanism by which MI alleviates aging remains unclear in *C. elegans*.

In this study, we are interested in whether MI can extend the lifespan and improve the health of *C. elegans* by attenuating the IIS pathway. We hypothesized that MI could alleviate aging in *C. elegans* by attenuating the IIS pathway when the interference of the osmotic pressure was excluded. It is known that the *C. elegans* can be cultured on the plate with the NGM containing tested agents or spreading tested agents on the top surface of the agar together with the *E. coli* OP50 [25,26]. The latter can exclude the interference of the osmotic pressure and be adopted in this study. We first confirmed the effects of MI on the longevity of the *C. elegans*, which were assessed according to the lifespan assay, as well as the evaluation of MI on the aging marker of the autofluorescence and two physiological indexes, i.e., pharyngeal pumping and body bends for heath. We then uncovered the loss-of-function mutants by assessing the effects of MI on the lifespan, physiological indexes, and autofluorescence using mutants, such as AKT-1 and DAF-16, to reveal the IIS pathway involved mechanism in the MI-ameliorating lifespan and health of the *C. elegans*. The AKT-1 is an AKT isoform in the *C. elegans* [5]. In addition, we also tested the nuclear localization of the DAF-16 to assess the activation of the DAF-16 by MI by using the MQD1543 strain of the *C. elegans*, in which the DAF-16 is tagged with the green fluorescent protein (GFP). Because the activated DAF-16 will translocate from the cytoplasm into the nucleus, thus, the nuclear localization assay can detect the activation of the DAF-16 [27]. The activation of the DAF-16 can be used to confirm the attenuation of the IIS pathway. We further intended to address the modulating ability of MI on the PI3K. However, to the best of our knowledge, there are no commercial antibodies against the *C. elegans*’ PI3K, AKT and phosphorylated AKT. Thus, we decided to use broad-species-specificity antibodies for detecting these proteins in this study. In addition, we used a commercial human PI3K activity kit to measure the inhibiting activity of MI on the PI3K activity in vitro.

## 2. Results

### 2.1. Effects of MI on the Lifespan in the C. elegans When the Osmotic Pressure Interference Was Excluded

Different dosages of MI (0, 25, 50, 100, and 200 nmol/plate) were spread onto the central surface of the NGM to test the effects of MI on the lifespan of *C. elegans*. We used these dosages of MI because we had previously tested the effect of MI on lifespan extension, and we found that 100 nmol/plate had the best effect on lifespan extension in *C. elegans*. Therefore, we performed the lifespan assay using several dosages of around 100 nmol/plate. The results showed that MI could extend the lifespan in a dose-dependent manner. Again, the MI at a dosage of 100 nmol/plate showed having the most effective in extending the lifespan of the *C. elegans*; the mean lifespan was increased by 22%, and both the median lifespan and maximum lifespan were increased by two and eight days (Figure 1; Table 1). Comparing the obtained results with the previous study [23], when they administered MI to the *C. elegans* by dissolving the MI in the NGM, they found that 1–700 mM MI could extend the lifespan of the *C. elegans*, and 500 mM of MI had the best effect, which extended the lifespan of the *C. elegans* by 31%. In this study, the efficacy was lower than the 31% achieved in the previous study with the 500 mM MI dissolved in the NGM [23]; however, this lower efficacy ruled out interference from the osmotic pressure.

### 2.2. Effects of MI on the Pharyngeal Pumping, Body Bend and Autofluorescence in the C. elegans When the Osmotic Pressure Interference Was Excluded

Different dosages of MI (0, 25, 50, 100, and 200 nmol/plate) were spread onto the central surface of NGM, and the effects of MI on the health indexes of pharyngeal pumping and body bend in the *C. elegans* are shown in Figure 2. The results showed that MI at doses ≤100 nmol/plate could significantly (*p* < 0.001) increase the pharyngeal pumping of the *C. elegans* (Figure 2a) and the body bends of the *C. elegans* in the swimming test (*p* < 0.001) (Figure 2b), suggesting that MI could promote the health of the *C. elegans* in a dose-dependent manner at the doses of MI ≤100 nmol/plate. Additionally, MI could significantly (*p* < 0.05) reduce the aging maker of autofluorescence of the *C. elegans* in a dose-dependent manner at doses of MI ≤100 nmol/plate (Figure 2c). However, when the MI dose was higher than 200 nmol/plate, the dose-dependent amelioration on the pharyngeal pump, body bend and autofluorescence disappeared.

### 2.3. Effects of MI on the Lifespan, Health Indexes and Autofluorescence of the AKT-1 Mutants

To reveal whether the functions of MI were lost in the AKT-1 mutants, the mutants were grown on plates with or without 100 nmol MI/plate. The results showed that MI could not extend the lifespan of AKT-1 mutants (Figure 3a and Table 2), demonstrating that the lifespan extension of the *C. elegans* by MI is AKT-1-dependent. Moreover, the increase in pharyngeal pumping and body bend and the reduction in autofluorescence induced by MI also disappeared in the AKT-1 mutants (Figure 3b–d), suggesting that the effects of MI on the health indexes and the aging marker depend on AKT-1.

### 2.4. Effects of MI on the Lifespan, Health Indexes and Autofluorescence of the DAF-16 Mutants

The results show in Figure 4a and Table 2 that MI cannot extend the lifespan of the DAF-16 mutants. The effects of MI to significantly increase the pharyngeal pumping and the body bends and reduce the autofluorescence of the *C. elegans* were also lost in the DAF-16 mutants (Figure 4b–d). These results suggested that MI alleviates aging in *C. elegans* through a DAF-16-dependent mechanism. 

### 2.5. Effects of MI on the Lifespan, Health Indexes and Autofluorescence of the DAF-18 Mutants When the Osmotic Pressure Interference Was Excluded

Shi et al. [23] demonstrated that the MI-alleviating aging in *C. elegans* was dependent on the DAF-18. Therefore, we further intended to confirm the DAF-18 dependency in the *C. elegans* when the interference of the osmotic pressure was excluded. As shown, the lifespan-extending effect of MI (Appendix A, and Table 2) and the effects of MI to significantly increase the pharyngeal pumping and the body bends and to reduce the autofluorescence of the *C. elegans* did disappear in the DAF-18 mutants (Appendix A). The results showed that the longevity effects of MI in the *C. elegans* depended on the DAF-18 when excluding the interference of the osmotic pressure.

### 2.6. Effects of MI on the Nuclear Localization of the DAF-16

We further measured the DAF-16 nuclear localization to reveal if the DAF-16 is activated by MI and is translocated into the nucleus. The representative photos of the DAF-16 distribution in the cytosolic localization, intermediate localization, and nuclear localization are shown in Figure 5a–c. Using the chi-squared test for the semi-quantitative data analysis, the results showed that the distribution of the DAF-16 was significantly different among the three groups (*p* < 0.05). The results showed that the nuclear localizations were 5.6%, 22.2%, and 36.7% in the control, 50, and 100 nmol/plate MI groups, respectively, while the cytosolic localizations were 72.2%, 46.7%, and 38.9% in the three groups (Figure 5d). The results demonstrated that MI could dose-dependently induce the DAF-16 nuclear translocation.

### 2.7. Effects of MI on the Expression of PI3K and Phosphorylation of AKT in the C. elegans and the Hs68 Cells

To assess the effect of MI on the expression of nematodes’ PI3K and the level of phosphorylated AKT (p-AKT), the *C. elegans* were grown in the NGM plate with a dosage of 100 nmol/plate of MI. The results showed that the expression of nematodes’ PI3K and the level of p-AKT significantly decreased (*p* < 0.05) (Appendix A). However, because of the limitation of the broad-species-specificity antibodies used, the band signal of the Western blotting obtained was not good enough. 

To confirm the effects of MI on the expression of the PI3K and phosphorylation of the AKT, we also used the human Hs68 cells to reveal the question. Hs68 is a normal skin fibroblast and has been used in our previous studies. The cells were cultured in Dulbecco’s Modified Eagle Medium (DMEM) with 5% CO_2_ at 37 °C. The detailed information for the Hs68 cells, including the culture conditions, is referred to in our previous paper [26]. For the Hs68 cells, we additionally used the LY-294002 as a control for the modulation of the PI3K by MI. When the Hs68 cells were grown in the medium with concentrations of 0.5 and 1 mM of MI, the results showed that the expression of PI3K decreased by the MI in a dose-dependent manner (Figure 6a,b). In contrast, the PI3K inhibitor, LY-294002, did not affect the PI3K expression. In addition, MI treatment of Hs68 cells at 0.5 and 1 mM decreased AKT phosphorylation, but not AKT expression, in a concentration-dependent manner (Figure 6a,b). LY-294002 showed to reduce the phosphorylation of AKT in a concentration-dependent manner. These results demonstrated that MI has the ability to downregulate the expression of PI3K and inactivate AKT protein. The used concentrations of both MI and LY-294002 had no cytotoxicity on the Hs68 cells.

### 2.8. Effects of MI on the PTEN Expressions in the Hs68 Cells

As shown in Figure 6a,b, Both MI and LY-294002 increased the expressions of PTEN in a dose-dependent manner. 

### 2.9. Effects of MI on the PI3K Activity In Vitro

Because LY-294002 could upregulate the expression of PTEN by inhibiting the PI3K activity, we were interested in whether MI acts similarly to LY-294002 to inhibit PI3K activity. We further tested the effect of MI on the PI3K activity; the results showed that MI could dose-dependently inhibit the activity of PI3K with an IC50 of 90.2 μM for p110α (Figure 7a) and 21.7 μM for p110β (Figure 7b).

### 2.10. Effects of MI on the PTEN Activities In Vitro

The effect of MI on the activity of PTEN was evaluated by using the commercial Malachite Green Phosphatase Assay Kit. The results showed that the concentrations of MI from 0 to 1000 μM all had no significant effect (*p* > 0.05) to potentiate the PTEN activity (Figure 8b), demonstrating that PTEN is not a direct molecule target of MI. 

## 3. Discussion

The purpose of this study was to explore the role of the IIS pathway in the MI-extending the lifespan and improving the health of the *C. elegans*. When excluding the interference of the osmotic pressure, the results showed that MI could still significantly extend the lifespan of *C. elegans*. The most effective dosage of MI (i.e., 100 nmol/plate) extended the mean lifespan of *C. elegans by* about 22%. MI also alleviated the aging marker of autofluorescence and promoted health by ameliorating the physiological indexes of the pharyngeal pumping and body bend. The results have confirmed the previous report that MI can alleviate the aging of *C. elegans* [23]. However, the lifespan-extending effects of MI disappeared in the DAF-16 and AKT-1 mutants. The ameliorated effects of MI on pharyngeal pumping, body bend, and autofluorescence also disappeared in both the DAF-16 and AKT-1 mutants. In addition, MI showed that the dose-dependently induced the nuclear translocation of the DAF-16. These results have demonstrated that MI could attenuate the IIS pathway, and the longevity effect of MI is AKT and DAF-16 dependent. In this study, we also found that MI could inhibit human PI3K activity in a dose-dependent manner. MI could also significantly downregulate the expression of PI3K and decrease the phosphorylation of AKT in the Hs68 cells. All of these results have demonstrated that MI can inhibit the PI3K activity and downregulate the PI3K expression, and the attenuation of the IIS pathway plays a crucial role in MI to alleviate aging in *C. elegans*. Importantly, to the best of our knowledge, this study is the first in the literature to demonstrate that MI is an inhibitor of PI3K.

It is known that the mechanisms of action of MI are typically involved in lipid signaling, osmolarity, glucose metabolism, and insulin-sensitizing properties [3,4]. In this study, we demonstrated a new mechanism of action of MI, i.e., MI is an inhibitor of PI3K. Importantly, this new mechanism could be related to the longevity effects of MI in *C. elegans*. Different from the PI3K inhibitor of the LY-294002, MI not only inhibited the PI3K activity but also downregulated the PI3K expression (LY-294002 can only inhibit the PI3K activity). The results have suggested that the downregulation of the PI3K by MI is not owing to the inhibition of the PI3K activity, and another unknown mechanism could be involved in the MI-induced PI3K expression. Interestingly, due to the pivotal role of the IIS pathway in longevity, the LY-294002 that inhibits the PI3K was proposed as a candidate as a CRM for alleviating aging [9]. In this study, we found that the property of the PI3K inhibition of MI has made the potential of MI for alleviating aging. The CRM potential of MI also merits further investigations. Moreover, both LY-294002 and MI can upregulate the PTEN expression, suggesting that PTEN is a downstream signal of PI3K. Indeed, PTEN has been demonstrated as a downstream signal of PI3K/AKT/FOXO [28,29]. Thus, MI and LY-294002 should inhibit PI3K so as to activate FOXO to upregulate the PTEN expression. The upregulation of PTEN also has supported that the longevity effect of MI is AKT and DAF-16 dependent in the *C. elegans*. Compared to the LY-294002, the advantage of MI as a PI3K inhibitor is that MI is a nutrient. The safety of nutrients is generally considered to be higher. In fact, human studies in the literature have demonstrated a high level of safety for oral supplementation with MI [10]. However, at doses higher than 200 nmol/plate MI, the improvement in the physiological indicators in *C. elegans* may be reduced, suggesting that excessive attenuation of the IIS pathway may have an adverse health effect. Human supplementation with MI still needs to follow the recommended dosage for the health effect.

The PI3K superfamily of enzymes is lipid kinases, which are composed of catalytic and regulatory subunits [30,31]. Based on the primary structure, regulation, and in vitro lipid substrate specificity, mammalian PI3K can be divided into three classes (i.e., I, II and III) [30]. Among them, class I PI3K was found to phosphorylate signaling lipid PIP2 to PIP3 and is highly related to cancer, diabetes, and aging [31]. Class I can further be divided into α, β, γ and δ by the sequence of the catalytic subunit. The α and β isoforms (molecular weight is 110) are expressed in all types of cells, but γ and δ are expressed in leucocytes. To the best of our knowledge, this study is the first one to report that MI can be an inhibitor of both p110α and p110β. Although this study did not measure the inhibitory effect of MI on the *C. elegans*’ PI3K activity, since PI3K is a superfamily, MI can inhibit the activity of human PI3K, suggesting the same inhibitory effect in the *C. elegans*. Furthermore, PINT has been discovered as a gene induced by overexpression of PTEN [32], but actually, PINT doesn’t directly interact with PTEN. Studies have demonstrated that PINT is a downstream molecule of DAF-16/FOXO [29,33,34]. The previous study reported that MI could upregulate the PINT expression to regulate mitophagy [23], which has supported that the longevity effect of MI required DAF-16 in the *C. elegans*. 

Based on the results obtained in this study, we propose a possible mechanism for the longevity effects of MI in the *C. elegans* in Figure 9. Because PI3K is a molecule target of MI, the inhibition of the PI3K activity by MI will attenuate the IIS pathway by decreasing the phosphorylation of the AKT to induce the activation of the DAF-16. It is known that the DAF-16 has a longevity effect [5,6]. Furthermore, an unknown mechanism induces the downregulation of the PI3K expression, which will further enhance the attenuation of the IIS pathway. Moreover, MI has been demonstrated to have a longevity effect via the DAF-18 in *C. elegans* [23]. The DAF-18/PTEN can attenuate the IIS pathway by the change of PIP3 to PIP2, as well as the DAF-18/PTEN that has been demonstrated as a downstream signal of the FOXO [28,29]. Thus, we proposed that the inhibition of PI3K activity by MI can upregulate the DAF-18 expression via the activation of the DAF-16, and the DAF-18 also enhances the attenuating effect on the IIS pathway. Finally, MI can have a significate longevity effect on *C. elegans*.

## 4. Materials and Methods

### 4.1. Materials

All chemicals used were of analytical grade. NaCl, KCl, NaOH, Na_2_HPO_4_, and HCl were purchased from Merck (Darmstadt, Germany). Myo-inositol, ampicillin sodium salt, fluorodeoxyuridine (FUdR), cholesterol, DL-dithiothreitol, sodium dodecyl sulfate, bromophenol blue and sodium azide were obtained from Sigma (St. Louis, MO, USA). Ethanol, methanol, glycerol, Tris-base, KH_2_PO_4_, MgSO_4_, and CaCl_2_ were purchased from J.T. Baker^®^ (Phillipsburg, NJ, USA). Tween 20 was purchased from Panreac (Barcelona, Espana). American bacteriological agar and yeast extract was obtained from Conda (Madrid, Spain). Vegetable peptone and tryptone were obtained from Fluka (Buchs, Switzerland). Ammonium persulphate, N, N, N, N-tetramethylethylenediamine (TEMED) were purchased from Bio-rad (Hercules, CA, USA). 30% Acrylamide/Bis solution was purchased from Apollo Scientific (Chesire, UK). PI3K (Cat No. sc-1637) and β-actin (sc-47778) antibodies were purchased from Santa Cruz Biotechnology (Santa Cruz, CA, USA). Antibodies to phosphorylated AKT (Cat No. GTX128414), AKT (GTX121937), and PTEN (GTX101025), and secondary antibodies to mouse (GTX213111-01) and rabbit (GTX213110-01) were obtained from GeneTex (Irvine, CA, USA). The wild-type *C. elegans* strain N2 was a gift from the *C. elegans* core facility (National Taiwan University, Taipei, Taiwan). The AKT-1, DAF-16, and DAF-18 mutants and the MQD1543 strain were obtained from the Caenorhabditis Genetics Center (University of Minnesota, Twin Cities, MN, USA). 

### 4.2. Handling Procedures for the C. elegans

Wild-type N2 *C. elegans* and various mutants were maintained and propagated on nematode growth medium (NGM) as described elsewhere [35]. In brief, nematodes were regularly grown at 20 °C in 10-cm plates in 20 mL/plate NGM, which had been spread with 10 mg (wet weight) of *E. coli* OP50. For the lifespan assay, Amp/FUdR plates were prepared using 5-cm plates by adding 10 mL/plate NGM containing 100 μg/mL ampicillin and 50 μM FUdR. In addition, 10 mg wet weight of OP50 bacteria in 200 μL of LB medium with or without MI were spread onto the NGM plate. After the surface LB medium was dried, the Amp/FUdR plates were then exposed to UV at two doses of 9999 × 100 mJ/cm^2^ to kill the bacteria on the surface of the plates. The preparation of NGM, LB medium and M9 buffer, and other procedures were performed as described previously [26]. 

### 4.3. Synchronization of the C. elegans

The procedures of nematode synchronization were based on the method proposed by Sutphin et al. [35]. Picked the appropriate number of adult nematodes cultured on the dead bacteria NGM plate to the new dead bacteria NGM plate. On the next day, the adults were taken away, and the eggs were left behind. Waiting until the eggs hatch and grow to L2 -L3 stage (approximately 1–2 days), the synchronization was completed. 

### 4.4. Assay of the Lifespan of the C. elegans

The lifespan of the nematodes was determined as described in the previous publication with slight modifications [35]. Synchronized larvae at the L1 stage were gently picked onto a 5-cm Amp/FUdR plate (30 worms/plate; 3 plates per every treatment, n = 90) containing various dosages of MI on the top surface of the plate and grown at 20 °C. The plates of all groups were replaced with new plates every 4 days, and the surviving nematodes were counted every two days. The worms were considered dead when they did not react to the touch of a platinum wire or lost their pharyngeal pumping. The survival percentage and average, median, and maximum lifespan values were obtained in SPSS, and the recorded survival percentage was plotted by Sigma Plot 8.0. In order to eliminate the interference of the osmotic pressure, we added an appropriate concentration of MI to the LB medium containing *E. coli* OP50 and spread 200 μL of the prepared MI-containing bacterial solution onto the surface of the NGM, as per our previous report [26]. Because the LB medium would dry out after spreading onto the surface of the NGM, thus, the dosage unit of nmol/plate (i.e., from 0 to 200 nmol/plate) was used rather than the molar concentration of MI [25,26].

### 4.5. Assay of the Pharyngeal Pumping in the C. elegans

Pharyngeal pumping was determined as described elsewhere [36,37]. Synchronized larvae are grown in 5-cm Amp/FUdR plates (30 worms/plate) containing various dosages of MI. The plates of all groups were replaced with new plates every 4 days, and the survival worms were left in the fresh plate. Ten nematodes for each group were randomly chosen to record pharyngeal pumping with a charge-coupled device (CCD) video camera under a microscope on the 4th^,^ 6th and 8th days of life. The number of pharyngeal pumps within 30 s was calculated based on the slow-motion playback.

### 4.6. Assay of the Body Bends in the C. elegans

Body bends were determined by the swimming test as described previously [23,36]. Synchronized larvae are grown in 5-cm Amp/FUdR plates (30 worms/plate) containing various dosages of MI. The plates of all groups were replaced with new plates every 4 days, and the survival worms were left in the fresh plate. Ten nematodes for each group were randomly picked up to determine the body bends on the 4th, 6th and 8th days of life. After crawling on an NGM plate without *E. coli* for 30 s to remove excess *E. coli* from the worms; each nematode was placed in a well of a 24-well plate containing 1 mL of M9 buffer. The motility of nematodes was recorded with a CCD video camera under a microscope, and the number of body bends within 30 s was calculated based on the slow-motion playback. Body bends were defined as the number of repeated twists at the center point of the nematode.

### 4.7. Determination of the Autofluorescence in the C. elegans

Autofluorescence was determined as described previously [38]. Synchronized larvae are grown in 5-cm Amp/FUdR plates (30 worms/plate) containing various dosages of myo-inositol. The plates of all groups were replaced with new plates every 4 days, and the survival worms were left in the fresh plate. Nematodes (*n* = 3) were picked onto a slide coated with 2% agar pad (1 worm/slide) on the 10th day of life. Nematodes were paralyzed with 20 mM sodium azide, and the fluorescence of the nematodes was imaged by a ZEISS Axio Imager A2 fluorescence microscope (Zeiss, Göttingen, Germany) with a Rhod filter. Red autofluorescence was quantified by the ImageJ image analysis software.

### 4.8. Distinguishing the Loss-of-Function Mutants

It is known that a mutation in DNA results in the decreased production of a protein or a protein with impaired functions, namely loss-of-function mutation. One of the powerful advantages of the use of the *C. elegans* model is using the mutants to reveal the signaling pathway of a certainly tested molecule. If a certain function, such as the lifespan-extending effect, disappears in a certain mutant, the result indicates that the signaling mechanism of the tested molecule is via the mutated protein. The AKT-1, DAF-16 and *DAF-18* mutants were used for the selection of loss-of-function mutants in this study. Except that the wild-type nematodes were replaced with mutants, all the procedures performed were the same as the above assay of the lifespan, pharyngeal pumping, body bend and autofluorescence of the *C. elegans*.

### 4.9. DAF-16 Nuclear Localization Assay in the C. elegans

The DAF-16 nuclear localization assay referred to the method used by Tao et al. [27]. After being treated with the tested compound for five days, the MQD1543 strain of nematodes was picked on a culture plate without *E. coli* with platinum wire to remove *E. coli* on the *C. elegans*. After placing them onto slides containing 2% agar, the nematodes were anesthetized with 20 mM sodium azide, and the DAF-16::GFP fluorescence was measured with a ZEISS Axio Imager A2 fluorescence microscope. The DAF-16 localization of each animal was scored as having cytosolic localization and nuclear localization when the localization is observed throughout the entire body from head to tail, or intermediate localization when there is a visible nuclear localization but one not as complete as nuclear [39,40]. The number of worms with each level of nuclear translocation was counted.

### 4.10. Western Blotting in the C. elegans and Hs68 Cells

Protein expression of nematodes’ PI3K, AKT, and p-AKT was measured by Western blotting. Worm lysate was prepared based on the method proposed by Jeong et al. [41]. Briefly, 10 mg (wet weight) of worms in a vial with the proper volume of the sample buffer (0.125 M Tris-Hcl (pH = 6.8), 4% SDS, 20% glycerol, 0.05% bromophenol blue and 10–20 mM DTT) were frozen and thawed for three times, repeatedly. 20 µL of worm lysate was resolved by SDS−PAGE and transferred onto a polyvinylidene fluoride (PVDF) membrane. After being blocked with phosphate buffer saline (PBS) containing 0.1% (*v*/*v*) Tween 20 (i.e., PBST) buffer containing 5% nonfat milk and 0.5% BSA in PBST, the membrane was washed three times with PBST buffer for 1 h and then incubated with different primary antibodies including PI3K, AKT, and p-AKT antibodies at 4 °C overnight. The membrane was incubated with a fluorescein-conjugated secondary antibody for 1 h and then detected with an ECL chemiluminescent detection kit (T-Pro Biotechnology, Taipei, Taiwan).

Protein expressions of the PI3K, AKT and PTEN, and p-AKT in the Hs68 cells were measured by Western blotting. Proper numbers of cells were plated onto a 100-mm dish. After various treatments at the indicated incubation time, cell lysates were collected in microcentrifuge tubes by using 400 μL of CytoBusterTM Protein Extraction Reagent (Novagen, Madison, WI, USA) containing 1% protease inhibitor cocktail (CalBiochem, Darmstadt, Germany) and 10 μM of TSA. Protein concentrations were determined using the BCA protein assay kit (Novagen, Madison, WI, USA). 15 μg of cell lysate were resolved by SDS-PAGE, transferred onto PVDF membrane (Bio-Red, Hercules, CA, USA), and immunoblotted for PI3K, AKT, p-AKT, PTEN and β-actin. Protein was visualized using the Trident femto Western HRP Substrate (GeneTex, Irvine, CA, USA). The protein expressions were quantified with Image J software. Relative expression of protein or phosphorylation of the AKT was calculated by the protein density or the ratio of p-AKT/AKT to control [42], respectively. 

### 4.11. Measurement of the Inhibition of MI on the PI3K Activity

The inhibition of MI on the human PI3K activity was measured by using a commercial kit of the PI3 Kinase Activity/Inhibitor Assay Kit (Catalog No. 17-493, Millipore, Darmstadt, Germany) with a protocol that followed the instructions. Briefly, pre-incubate the kinases (including p110α and p110β) and the different concentrations of MI (from 0–1000 μM) for 10 min. Add 5 μL/well of the 5X kinase reaction buffer, 5 μL/well of the PIP2 substrate, and distilled H_2_O to each well to make up for a final 25 μL/well. Add 25 μL/well 1XTBS to the buffer control wells. After incubation at room temperature (RT) for one hour, all of the wells were added 25 μL/well of the biotinylated-PIP3/EDTA working solution, excluding the buffer control wells. Then, 50 μL/well of the general receptor of phosphoinositide’s 1 (GRP1) working solution was added to all the wells. After incubation at RT for one hour, the wells were washed with 200 μL/well 1XTBST four times. Add 50 μL/well to the streptavidin-horseradish peroxidase working solution, and incubate at RT for one hour. After washing, the substrate TMB (Catalog 90348) 100 μL per well was added and developed in the dark for 5–20 min. The reaction was stopped by adding 100 μL of the stop solution (Part No. 2007598) per well. The absorbance was read at 450 nm. The relative % to B-PIP3 was calculated by (the A450 of samples including buffer, kinase and MI/A450 of the biotinylated-PIP3 average) × 100. The relative PI3K activity (% of control) was further obtained by calculating (the close % to B-PIP3 of the buffer control/the relative % of B-PIP3 of the kinases with MI) × 100.

### 4.12. In Vitro PTEN Activity Assay

The phosphatase activity of the PTEN was determined using the Malachite Green Assay Kit (Echelon Biosciences, Salt Lake, UT, USA) [43]. Briefly, the recombinant PTEN enzyme (Echelon Biosciences) was reconstituted in distilled water to yield a 50 μg/mL concentrate. Phosphatidylinositol 3,4,5-trisphosphate diC8 (PIP3) was purchased from Echelon Biosciences. Each reaction (final volume of 25 μL) contained 75 ng PTEN enzyme and 3 nmol of PIP3 in the PTEN reaction buffer (25 mM Tris–HCl pH 7.4, 140 mM NaCl, 2.7 mM KCl, 10 mM DTE). After incubation at 37 °C for 60 min, 100 μL of malachite green solution (Malachite Green Phosphatase Assay Kit, Echelon K-1500) was added. After 20 min of development of the color at room temperature, the absorbance at 620 nm was measured. The phosphate standards (Echelon K-1500 kit) were used to create the standard curve to quantify phosphate production. The relative phosphate amount to the control (i.e., 0 μM of MI) was calculated and represented as the PTEN activity.

### 4.13. Statistical Analysis

Data were analyzed using Student’s t-test or analysis of variance (ANOVA), followed by Duncan’s test for group mean comparisons using the SPSS v.14.0 software (SPSS, Inc., Chicago, IL, USA). The differences in the lifespan of the nematodes between the groups were analyzed using the log-rank test. Since the log-rank test is suitable for comparing the survival distributions of two groups, the *p* values were corrected by the Bonferroni correction method when the number of groups was ≥3. The pharyngeal pumping and body bend were analyzed using a simple regression analysis method. The DAF-16 nuclear localization assay was analyzed using the chi-squared test. *p* values less than 0.05 were considered statistically significant. 

## 5. Conclusions

In summary, we confirmed that MI could extend the lifespan of *C. elegans*, improve health indexes, including pharyngeal pumping and body bends, and ameliorate the aging marker of the autofluorescence when the osmotic pressure interference was excluded. In addition, the longevity effects of MI in the *C. elegans* depend on the AKT-1 and DAF-16. The IIS pathway is significantly attenuated to induce the nuclear translocation of DAF-16 by the treatment of MI. We further found that MI can dose-dependently inhibit the PI3K activity in vitro and downregulates the expression of PI3K, and upregulates of the expression of PTEN in human Hs68 cells. Based on these results, we proposed that MI can modulate the PI3K activity by including direct activity inhibition and the downregulation of protein expression, which will attenuate the IIS pathway so as to activate DAF-16. The activation of the DAF-16 will further upregulate the expression of the DAF-18 to enhance the IIS pathway-attenuating effect and, finally, has a significate longevity effect on MI.

## Figures and Tables

**Figure 1 ijms-24-06194-f001:**
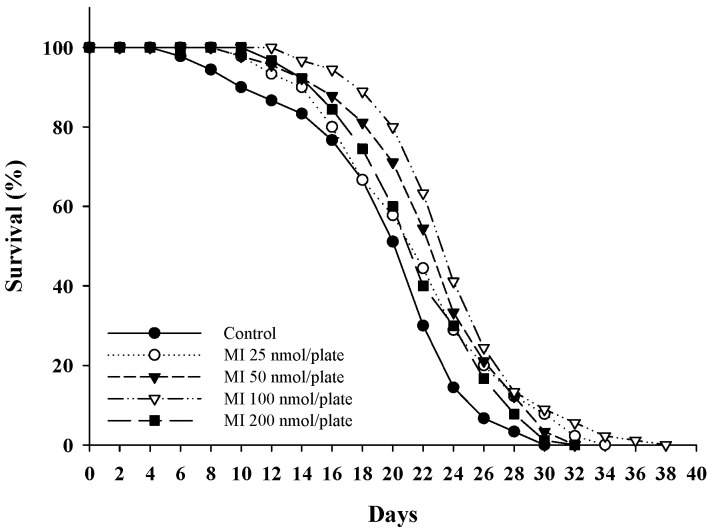
Effects of the myo-inositol (MI) on the lifespan of wild-type *C. elegans*. The nematodes (90 worms per group) were grown on plates containing different dosages of MI (0, 25, 50, 100, and 200 nmol/plate) as described in Methods. The results are from a representative experiment out of two independent experiments.

**Figure 2 ijms-24-06194-f002:**
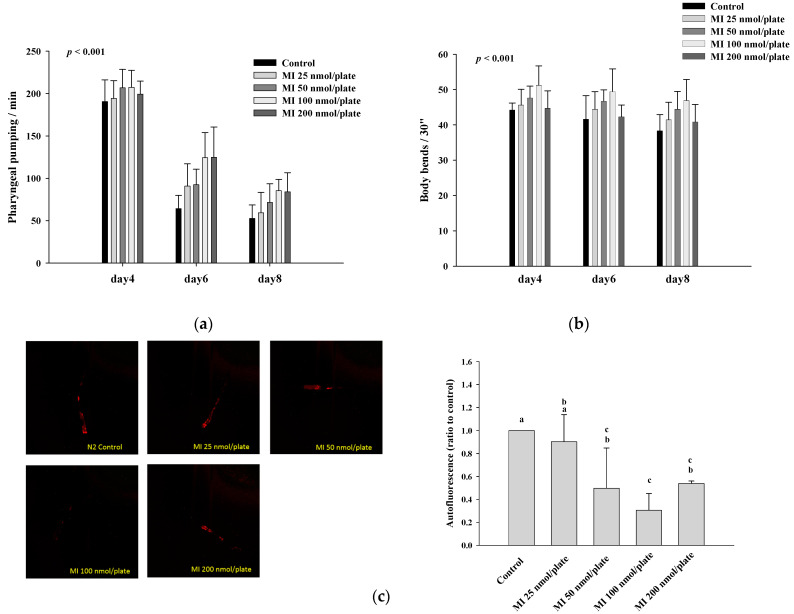
Effects of the myo-inositol (MI) on the pharyngeal pumping, body bends, and autofluorescence of wild-type *C. elegans*. The nematodes were grown on plates with different dosages of MI (0, 25, 50, 100, and 200 nmol/plate). On the fourth, sixth and eighth days, and the 10th day of life, (**a**) the pharyngeal pumping (*n* = 10), (**b**) body bends (*n* = 10), and (**c**) autofluorescence (left: photos; right: quantitative results, *n* = 3) were determined as described in Methods. For the results of pharyngeal pumping and body bends, the data were analyzed by the simple regression analysis method, but the data of 200 nmol/plate MI were excluded. *p* values are shown in the figures, or values (mean ± SD) not sharing a common letter are significantly different (*p* < 0.05). The results are from a representative experiment out of two independent experiments.

**Figure 3 ijms-24-06194-f003:**
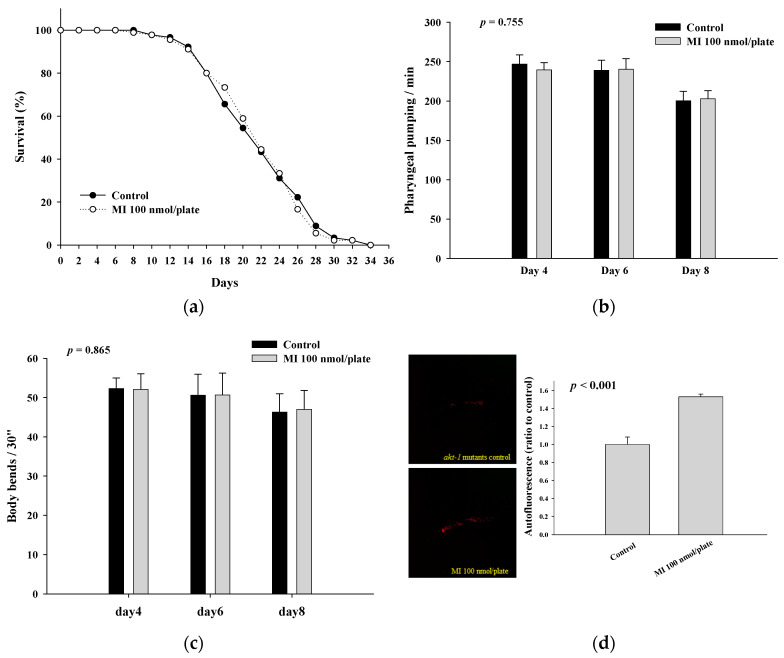
Effects of the myo-inositol (MI) on the lifespan and health of the AKT-1 mutants. (**a**) For the lifespan assay, various mutants (90 worms per group) were grown on plates containing with or without 100 nmol/plate of MI. The surviving worms were counted every two days. The surviving percentages were obtained, and the results were analyzed by the log-rank test with the SPSS software. For the health analysis, the nematodes were grown on plates containing with or without 100 nmol/plate of MI. (**b**) The pharyngeal pumping (*n* = 10), (**c**) body bends (*n* = 10), and (**d**) autofluorescence (left: photos; right: quantitative results, *n* = 3) were determined as described previously. Values are presented as mean ± SD, and *p* values are shown in the figures. The results are from a representative experiment out of three independent experiments.

**Figure 4 ijms-24-06194-f004:**
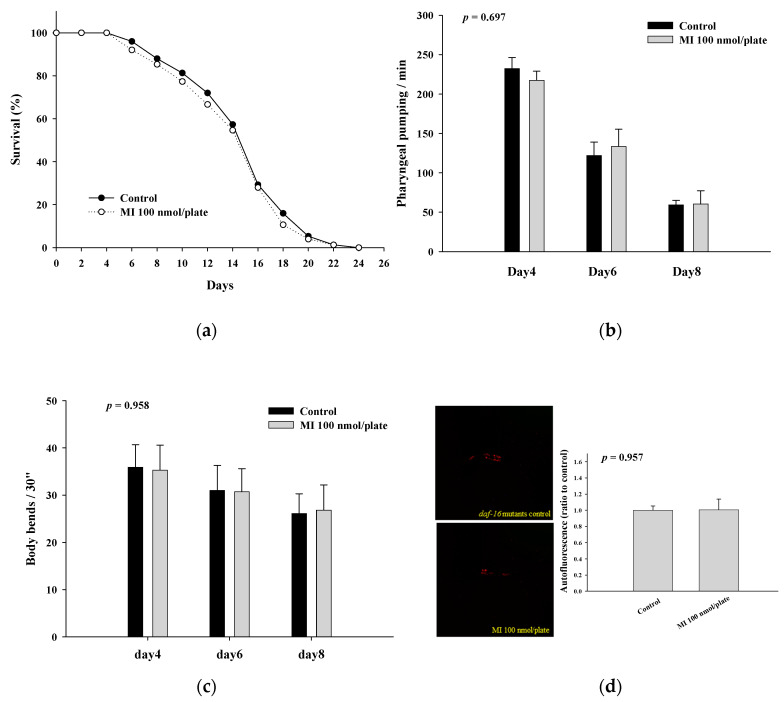
Effects of the myo-inositol (MI) on the lifespan and health of the DAF-16 mutants. (**a**) For the lifespan assay, various mutants (90 worms per group) were grown on plates containing with or without 100 nmol/plate of MI. The surviving worms were counted every two days. The surviving percentages were obtained, and the results were analyzed by the log-rank test with the SPSS software. For the health analysis, the nematodes were grown on plates containing with or without 100 nmol/plate of MI. (**b**) The pharyngeal pumping (*n* = 10), (**c**) body bends (*n* = 10), and (**d**) autofluorescence (left: photos; right: quantitative results, *n* = 3) were determined as described previously. Values are presented as mean ± SD, and *p* values are shown in the figures. The results are from a representative experiment out of three independent experiments.

**Figure 5 ijms-24-06194-f005:**
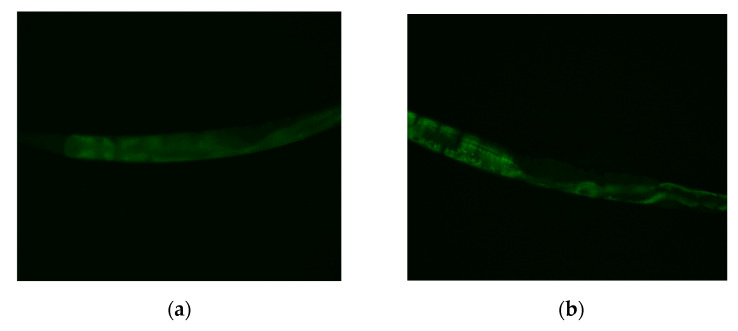
Effects the myo-inositol (MI) on the DAF-16 nuclear localization. After the MQD1543 strain of nematodes was grown on plates with different dosages of MI (0, 50 and 100 nmol/plate) for five days, the DAF-16:GFP fluorescence was measured as described in Methods. The representative photos at 200× magnification of (**a**) cytosolic localization, (**b**) intermediate localization, (**c**) nuclear localization, and (**d**) the semi-quantitative results of the DAF-16 distribution are shown. The data are from three independent experiments using 30 nematodes per group.

**Figure 6 ijms-24-06194-f006:**
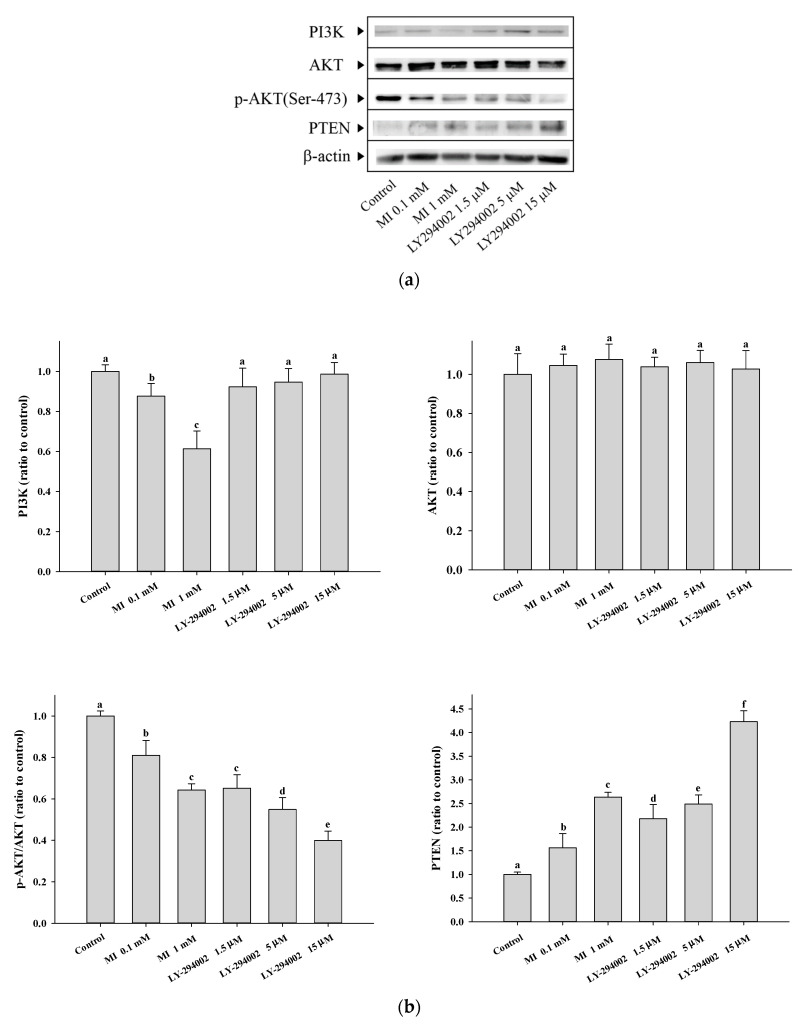
Effects of the myo-inositol (MI) on the expressions of PI3K, AKT, and PTEN, and the phosphorylation of AKT in Hs68 cells. After being treated with various concentrations of MI and LY294002 for five days, the cell lysates were obtained. The protein expressions or AKT phosphorylation level in the cell lysates were detected by the Western blot method as described in Methods. (**a**) Western blot Images; (**b**) Western blot quantification results (*n* = 3 independent experiments). Values (mean ± SD) not sharing a common letter are significantly different (*p* < 0.05).

**Figure 7 ijms-24-06194-f007:**
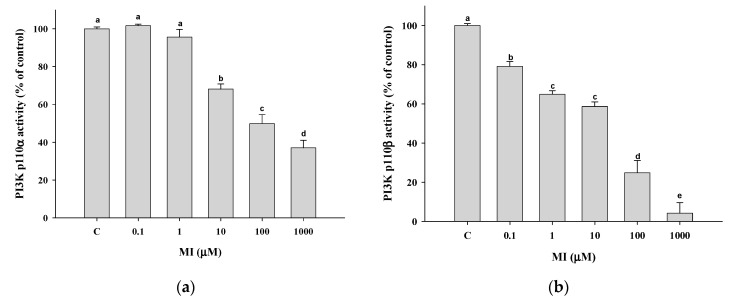
Effects of the myo-inositol (MI) on the phosphoinositide 3-kinase (PI3K) activity. Two isoforms of the PI3K enzyme, (**a**) p110α and (**b**) p110β, were incubated with different concentrations of MI. The PI3K activity was detected by a commercial PI3K kit as described in Methods. The inhibition percentage of MI on the PI3K isozymes is expressed as the mean ± SD (*n* = 3 independent experiments); Values not sharing a common letter are significantly different (*p* < 0.05).

**Figure 8 ijms-24-06194-f008:**
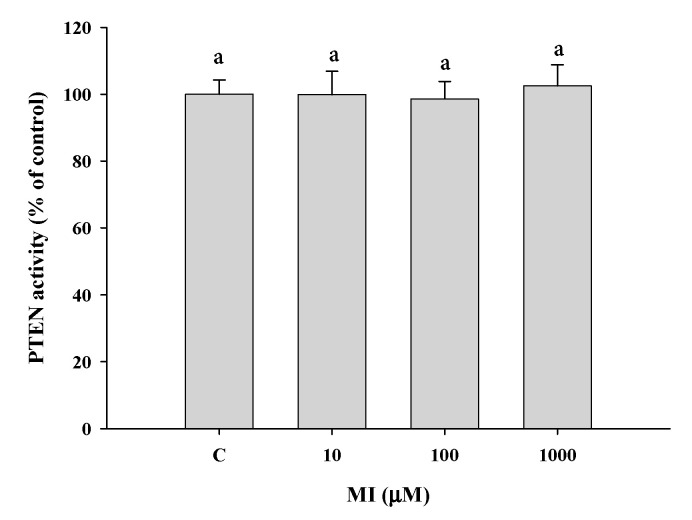
Effects of MI on the PTEN activity in vitro. The PTEN enzyme was incubated with different concentrations of MI (0–1000 μM). The PTEN activity was detected by a commercial Malachite Green Assay Kit as described in Methods. The relative PTEN activities are expressed as the mean ± SD (*n* = 3 independent experiments); Values not sharing a common letter are significantly different (*p* < 0.05).

**Figure 9 ijms-24-06194-f009:**
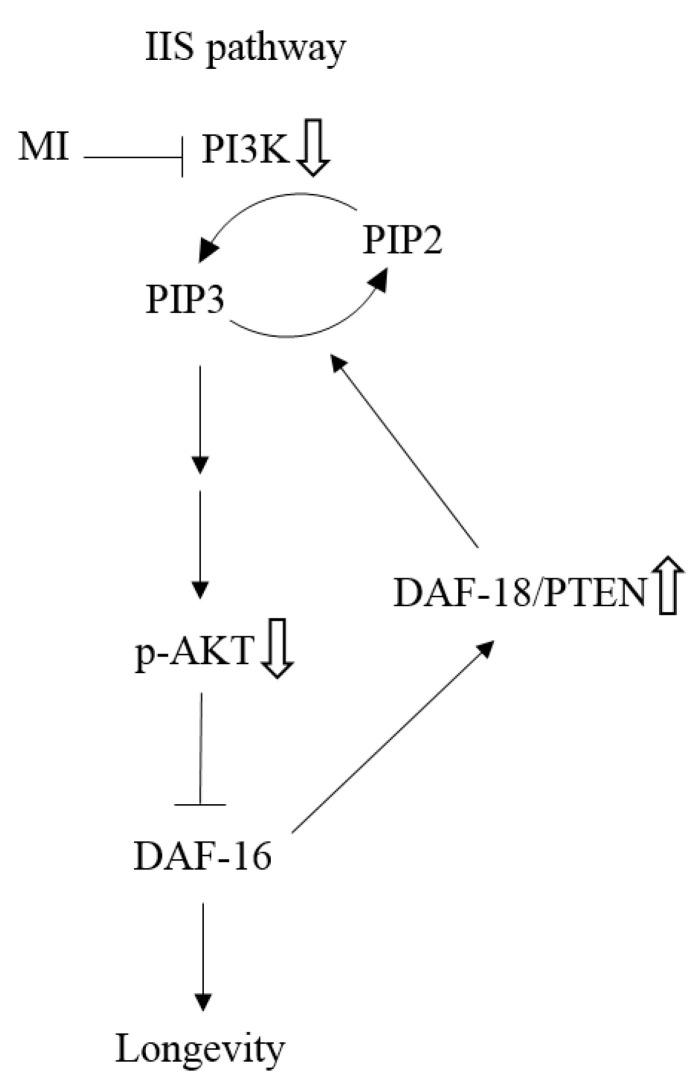
The schematic presentation of the mechanisms by which MI inhibits the signaling of insulin/insulin-like growth factor 1 (IIS) pathway. The symbol 
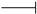
 as: “The inhibition step of MI or the inactivation step in the signaling pathway”, the symbol ⇩ as “The detected downregulation of protein expressions or protein phosphorylation by MI”, and ⇧ as “The detected upregulation of protein expression by MI” in this study. NB: the references are indicated in square brackets.

**Table 1 ijms-24-06194-t001:** Effect of myo-inositol (MI) on the lifespan of the *C. elegans*.

	Mean Lifespan (Day)	Median Lifespan (Day)	Maximum Lifespan (Day)	*p* Value *
Control	20.0 ± 5.6	22	30	
MI 25 nmol/plate	22.0 ± 5.7	22	34	0.028
MI 50 nmol/plate	23.0 ± 5.0	24	32	<0.001
MI 100 nmol/plate	24.4 ± 4.7	24	38	<0.001
MI 200 nmol/plate	22.1 ± 4.7	22	32	0.076

* *p* values were obtained when compared to the control group by log-rank test and were corrected by Bonferroni correction. Values of the mean lifespan (n = 90 for each group) are expressed as mean ± SD.

**Table 2 ijms-24-06194-t002:** Effect of myo-inositol (MI) on the lifespan of the mutants.

Mutants	Mean Lifespan (Day)	Median Lifespan (Day)	Maximum Lifespan (Day)	*p* Value *
**AKT-1** Control MI	21.9 ± 5.4	22	34	
	22.0 ± 5.3	22	36	0.921
**DAF-16** Control MI	14.9 ± 4.2	16	24	
	14.4 ± 4.3	16	24	0.517
**DAF-18** Control MI	14.2 ± 3.4	14	22	
	13.4 ± 3.4	14	20	0.121

* *p* values were obtained when compared to the control group by log-rank test. Values of the mean lifespan (n = 90 for each group) are expressed as mean ± SD. The dose of MI is 100 nmol/plate.

## Data Availability

Data can be provided upon request.

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
