# Peer review of "The Attenuation of Insulin/IGF-1 Signaling Pathway Plays a Crucial Role in the Myo-Inositol-Alleviated Aging in Caenorhabditis elegans"

_ijms, 2023, doi:10.3390/ijms24076194_

Round 1
Reviewer 1 Report
In this study, the authors have examined the mechanisms underlying the role of myo-inositol on increased lifespan of C. elegans. The comments regarding thsi manuscript are presented below:
1. The effect of increasing longevity observed with the highest dose of myo inositol was 2-3 days; what is the actual significance of such a modest increase in real life, especially in the context of human health and/or lifespan? Is there any evidence for myo-inositol increasing quality/ quantity of human life? These issues must be disucssed in the Introduction to justify this study.
2. The authors should choose a different abbreviation for myo-inositol as MI is the standard term used for myocardial infarction in bioscience/medical papers.
3. What is the effect of pharmacological inhibitor of PI3K upon the lifespan of wild-type as well as mutant C. elegans, in presene and/or absence of myo-inositol? These data would help establish teh critical role for I3K Pin this process.
4. The data from daf-18 mutant (Fig 8) should have been presented immediately following the data on daf-16 mutant.
5. The number of independent experiments shoudl be mentioned in each figure legend.
6. For western blotting, the primary and secondary antibodies used (Source, and species for all antibodies, clone for Primary antibodies) must be provided in the Methods section. Otehrwise, it could be impossible for another researcher to replicate and/or confirm these findings.
7. It is surprising that teh authors were unable to determine phosphorylation of Akt in C. elegans cells. A better justification is needed.
8. The biggest problem with this study is the sudden (and poorly explained jump from C. elegans to a human fibroblast cell line, Hs68). at the very least, a strong rationale should be provided for choosing this particular cell line, and greater detals provided (about its source, passage number, culture conditions morphology etc.) in the Methods section.
9. Finally, it is extremely concerning that up to 5000-fold higher concentrations (1 mM) of myo-inositol were used in cell culture of Hs68 as compared to those used (200 nM) in C. elegans studies. While low micromolar levels of reagents are often used in cell culture studies, these millimolar levels ar ealmost unheard of , and raise the probability of off-target and cytotoxic effects. The authors are strongly advised to: (a) validate their cell culture findings using a different human cell line, (b) use a wide range of myo-inositol concentrations in both cell lines, and (c) provide data on cell survival studies
Reviewer 2 Report
The Yang et al., 2023 manuscript ijms-2256222 addresses the crucial role of insulin/IGF-1 signaling pathway in the myo-inositol-alleviated aging in C elegans. A search on Pubmed.gov for the terms "Insulin" and "myo-inositol" and "Aging" keywords resulted in 8 hits that depicts the novelty of this study.
There are many queries and suggestion which makes this manuscript more representable to be publish.
Queries:
1. The authors have shown the experimental evidence for DAF-16 nuclear translocation, the imaging can be performed again to take smaller view showing full worm picture as well as the Bigger fluorescence images. Can you make a new image with better resolution?
2. It would be great to see the expression of insulin receptor in the MI treated cells?
3. The authors have treated the cells with various concentrations of MI and 211 LY294002 for five days. How is the survival ratio and cell viability pattern after this treatment?
4. The representation of phosphorylation of AKT in this study is not the correct way to represent the change and graph. The authors must show the ratio of pAKT/AKT has to be represented (Follow the calculation from this article PMID: 34303021, Kindly cite the study)?
5. Do the authors found any changes in mitochondrial biogenesis genes and ROS level in this study?
6. With increasing age, the expression of Insulin receptor got significantly decreased as showed by various studies in various organs. How can you justify?
7. What is the criteria to select the doses for the experiment in both human cells and worms?
Reviewer 3 Report
Remarks to the Author:
Nae-Cherng Yang and colleagues explored the crucial role the attenuation of insulin / IGF-1 signaling pathway in the myo-inositol-alleviated aging in Caenorhabditis elegans.
This topic is meaningful. But there are still some things that need to be revised carefully. Some conclusions were not supported by enough experimental data to elucidate the mechanism clearly, and the result analysis and discussion were insufficient.
Overall, I would support the publication of this study once the authors have addressed a series of changes.
My comments:
1. The logic of introduction needs to be reorganized. The introduction should include the research background, research purpose and significance, and should not involve the specific experimental steps.
2. What is the reference basis for the dose of MI (0, 25, 50, 100, and 200 nmol/plate)?
3. What are the possible reasons for the different trends of pharyngeal pumping, body bends and autofluorescence in MI 200 nmol/plate and MI≤100 nmol/plate? It should be discussed.
4. The result description is not detailed enough, as shown in Figure 2.
5. The statistical diagram of fluorescence results should be added in Figure 5.
6. In this manuscript, the author only explained that MI can regulate the life span of nematode through daf-16 and MI can regulate the longevity of C. elegans through daf-18, but did not do relevant experiments to prove that MI can regulate the life span of nematode through daf-16-daf-8 signal cascade directly.
Round 2
Reviewer 1 Report
The question regarding the effects of a pharmacological inhibitor of PI3K upon lifespan in both wildtype and mutant C. elegans, in the presence or absence of myo-inositol has not been addressed adequately by the authors. This is a critical experiment whose results would demonstrate the involvement (or not) of PI3K upon the observed actions of myo-inositol which is a central theme of this study.
Reviewer 2 Report
The authors have addressed the comments and queries raised by me which helped the manuscript to be ready for publication in the present form.
Reviewer 3 Report
The author and colleagues have answered and revised all the questions raised. Therefore, I support the publication of this study.
Round 3
Reviewer 1 Report
No further comments